# POSNOC—POsitive Sentinel NOde: adjuvant therapy alone versus adjuvant therapy plus Clearance or axillary radiotherapy: a randomised controlled trial of axillary treatment in women with early-stage breast cancer who have metastases in one or two sentinel nodes

Amit Goyal ,[1] G Bruce Mann,[2,3] Lesley Fallowfield,[4] Lelia Duley,[5] Malcolm Reed,[6] David Dodwell,[7] Robert E Coleman,[8] Apostolos Fakis,[9] Robert Newcombe,[10] Valerie Jenkins,[4] Diane Whitham,[5] Margaret Childs,[5] David Whynes,[11] Vaughan Keeley,[12] Ian Ellis,[13] Patricia Fairbrother,[14] Shabina Sadiq,[5] Kathryn Monson,[4] Alan Montgomery,[5] Wei Tan,[5] Luke Vale,[15] Tara Homer,[15] Heath Badger,[3] Rachel Helen Haines ,[5] Mickey Lewis,[5] Daniel Megias,[16] Zohal Nabi,[16] Preetinder Singh,[16] Andrei Caraman,[16] Elizabeth Miles,[16] on behalf of the POSNOC Trialists

For numbered affiliations see end of article.

**Correspondence to**
Associate Professor Amit Goyal;
amit.goyal@nhs.net

## ABSTRACT

**Introduction** ACOSOG-Z0011(Z11) trial showed that axillary node clearance (ANC) may be omitted in women with ≤2 positive nodes undergoing breast conserving surgery (BCS) and whole breast radiotherapy (RT). A confirmatory study is needed to clarify the role of axillary treatment in women with ≤2 macrometastases undergoing BCS and groups that were not included in Z11 for example, mastectomy and those with microscopic extranodal invasion. The primary objective of POsitive Sentinel NOde: adjuvant therapy alone versus adjuvant therapy plus Clearance or axillary radiotherapy (POSNOC) is to evaluate whether for women with breast cancer and 1 or 2 macrometastases, adjuvant therapy alone is non-inferior to adjuvant therapy plus axillary treatment, in terms of 5-year axillary recurrence.

**Methods and analysis** POSNOC is a pragmatic, multicentre, non-inferiority, international trial with participants randomised in a 1:1 ratio. Women are eligible if they have T1/T2, unifocal or multifocal invasive breast cancer, and 1 or 2 macrometastases at sentinel node biopsy, with or without extranodal extension. In the intervention group women receive adjuvant therapy alone, in the standard care group they receive ANC or axillary RT. In both groups women receive adjuvant therapy, according to local guidelines. This includes systemic therapy and, if indicated, RT to breast or chest wall. The UK Radiotherapy Trials Quality Assurance Group manages the in-built radiotherapy quality assurance programme. Primary endpoint is 5-year axillary recurrence. Secondary outcomes are arm morbidity assessed by Lymphoedema and Breast Cancer Questionnaire and QuickDASH questionnaires; quality of life and anxiety as assessed with

### Strengths and limitations of this study

► POsitive Sentinel NOde: adjuvant therapy alone versus adjuvant therapy plus Clearance or axillary radiotherapy (POSNOC) includes women undergoing mastectomy and those with extranodal invasion.

► Only women with 1 or 2 macrometastases are eligible for randomisation.

► POSNOC has an in-built radiotherapy quality assurance programme co-ordinated by the UK Radiotherapy Trials Quality Assurance Group.

► POSNOC is an international trial—UK, Australia and New Zealand.

FACT B+4 and State/Trait Anxiety Inventory questionnaires, respectively; other oncological outcomes; economic evaluation using EQ-5D-5L. Target sample size is 1900. Primary analysis is per protocol. Recruitment started on 1 August 2014 and as of 9 June 2021, 1866 participants have been randomised.

**Ethics and dissemination** Protocol was approved by the National Research Ethics Service Committee East Midlands—Nottingham 2 (REC reference: 13/EM/0459). Results will be submitted for publication in peer-reviewed journals.

**Trial registration number** ISRCTN54765244; NCT0240168Cite Now

## INTRODUCTION

Currently, women with early breast cancer and 1 or 2 macrometastases in sentinel lymph

nodes (SNs) generally undergo some form of axillary treatment, either surgical axillary node clearance (ANC) or radiotherapy to the axilla (ART). The use of axillary treatment is based on the assumption that it reduces the risk of axillary recurrence and might improve survival. However, axillary treatment damages lymphatic drainage from the arm and women can subsequently develop lymphoedema, restricted shoulder movement, pain, numbness, and other sensory problems. These adverse effects interfere with daily activities, are distressing, impair quality of life (QoL) and are costly to the NHS (National Health Service) in terms of rehabilitative treatments, as they are often permanent and symptom relief is difficult.

Axillary treatment may no longer be necessary for this group of women, as they usually have a low axillary tumour burden and current systemic therapy is effective at preventing breast and axillary recurrence.[1] Also, sentinel node biopsy has already removed the lymph nodes most likely to have metastasis.[2] Moreover, when adjuvant therapy includes radiotherapy (RT) to the breast or chest wall, the lower axilla may be treated as it is often included in the tangential irradiation field, and some lower level axillary nodes may be removed at mastectomy.[3]

This hypothesis is supported by the Z11 trial[4] that showed that ANC may be omitted in women with ≤2 positive nodes undergoing breast conserving surgery (BCS) and receiving whole breast RT. Axillary recurrence was low, and there were no clear differences between the two groups (ANC 0.5% vs no ANC 0.9%) at 6.3 years. Z11 has not had the desired impact on change of clinical practice because of several limitations. The study did not meet its recruitment goal, approximately 40% of participants had micrometastases, many were lost to follow-up (19.4%), and there was a lack of radiation therapy quality assurance. Most US radiation oncologists continue to treat the undissected axilla in women with micro or macro metastases with ART.[5] Therefore, it appears that clinicians in the US are interpreting Z11 in line with AMAROS (After Mapping of the Axilla: Radiotherapy Or Surgery?) study which showed that ART is a less morbid alterative to axillary lymph node dissection in women with low volume nodal disease[6] rather than omitting axillary treatment. Furthermore, the results are not applicable to subgroups that were not included such as those with mastectomy and microscopic extranodal invasion.

## OBJECTIVE

The primary objective of the POsitive Sentinel NOde: adjuvant therapy alone versus adjuvant therapy plus Clearance or axillary radiotherapy (POSNOC) trial is to evaluate whether, for women with early breast cancer and one or two nodes with macrometastases, adjuvant therapy alone is non-inferior to adjuvant therapy plus axillary treatment (ANC or ART), in terms of 5-year axillary recurrence.

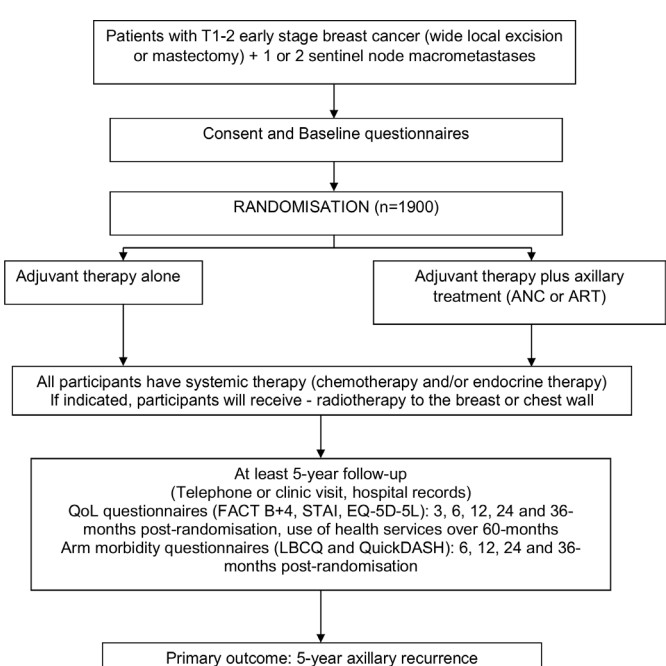

**Figure 1** Trial flow chart. ANC, axillary node clearance; ART axillary radiotherapy; FACT B+4, Functional Assessment of Cancer Therapy—Breast +4; STAI, State/Trait Anxiety Inventory; LBCQ, Lymphoedema and Breast Cancer Questionnaire; QuickDASH, disabilities of the arm, shoulder and hand questionnaire.

## METHODS AND ANALYSIS
### Design

POSNOC is a pragmatic randomised, multicentre, non-inferiority, international (UK, Australia, New Zealand) trial. Target sample size is 1900 participants, with 5-year follow-up. Flow of participants through the trial is summarised in figure 1.

### Randomisation

Participants are allocated via a secure internet-based system, using minimisation that is weighted toward minimising imbalance between trial arms with probability 0.8. Allocation is stratified by recruiting site and minimised by the following variables: age (<50, ≥50 years); type of breast surgery (BCS, mastectomy); oestrogen receptor status (ER) (positive, negative); number of positive nodes (1, 2); intraoperative sentinel assessment using OSNA (One-step nucleic acid amplification) (yes, no).

### Target population

Women with unifocal or multifocal invasive breast cancer with the largest lesion ≤5 cm, who have 1 or 2 sentinel nodes with macrometastases (>2 mm), with or without extranodal invasion.

Women are usually screened for eligibility after their initial breast surgery and sentinel node biopsy. In hospitals where sentinel node histology is assessed intra-operatively, women are asked to give their consent preoperatively and then randomised intra-operatively if their sentinel nodes are found to be positive.

## Inclusion criteria

Women are eligible for inclusion if all of the following criteria apply:

► 18 years or older.
► Unifocal or multifocal invasive tumour ≤5 cm in its largest dimension, measured pathologically or largest invasive tumour diameter on radiology for women who are randomised intra-operatively or who undergo sentinel node biopsy before neoadjuvant therapy (tumour size is based only on the single largest tumour).
► At sentinel node biopsy have 1 or 2 nodes with macrometastases (tumour deposit >2.0 mm in largest dimension or defined as macrometastases on molecular assay).
► Fit for axillary treatment and adjuvant therapy.
► Have given informed consent.

## Exclusion criteria

Women are excluded if they have any of the following:

► bilateral invasive breast cancer,
► more than 2 nodes with macrometastases,
► neoadjuvant therapy for breast cancer except
  – if sentinel node biopsy performed prior to neoadjuvant therapy in women with early breast cancer,
  – short duration of neoadjuvant endocrine therapy (up to 3 months),
► previous axillary surgery on the same body side as the scheduled sentinel node biopsy,
► not planned to receive adjuvant systemic therapy,
► previous cancer less than 5 years previously or concomitant malignancy except
  – basal or squamous cell carcinoma of the skin,
  – in situ carcinoma of the cervix,
  – in situ melanoma,
  – contralateral or ipsilateral in situ breast cancer.

## Trial treatment arms

All participants receive adjuvant therapy according to local guidelines. This includes chemotherapy and/or endocrine therapy for all women, and RT to breast or chest wall if indicated. Human epidermal growth factor receptor 2 targeted treatment is administered when indicated.

Trial participants are allocated to one of the following two groups:

1. Adjuvant therapy alone (intervention group)
   See above for adjuvant therapy. Axillary and supraclavicular fossa RT is not allowed.
2. Adjuvant therapy plus axillary treatment (standard care group)
   See above for adjuvant therapy. Axillary treatment can be ANC or axillary RT as per local guidelines.

## Outcomes

### Primary outcome

The primary outcome is 5-year axillary recurrence defined as pathologically (cytology or biopsy) and/or radiologically confirmed recurrence in lymph nodes draining the primary tumour site, i.e. nodes in the ipsilateral axilla, infraclavicular fossa, supraclavicular fossa and interpectoral area.

### Secondary outcomes

Secondary outcomes and timepoints of assessment are summarised in table 1.

► Arm morbidity assessed by (1) Lymphoedema and Breast Cancer Questionnaire[7]—the definition of lymphoedema is 'yes' to the two questions—'heaviness during the past year' and 'swelling now'; (2) QuickDASH questionnaire[8]—shoulder, arm and hand disability is defined as a change from baseline in the QuickDASH score of at least 14 points.
► QoL assessed by Functional Assessment of Cancer Therapy-Breast+4 (FACT B+4) questionnaire.[9] The total FACT B+4 score reflects global QoL and comprises physical (7 items), functional (7 items), social (7 items) and emotional well-being (6 items) plus the breast cancer concerns and arm morbidity scale (14 items) (total of 41 items; total score ranges from 0 to 164). The 5 item arm morbidity subscale score comprises the sum of the scores from items B3, B10, B11, B12, B13 (range 0–20) and will be analysed separately. Analyses will include the proportion of participants in the two allocated treatment groups reporting 'somewhat', 'quite a bit' and 'very much' for each of the 5 items.
► Anxiety assessed by using the Spielberger State/Trait Anxiety Inventory (STAI).[10] The STAI consists of 2 questionnaires with 20 items. It assesses anxiety proneness (Trait) and the current state of anxiety or anxiety change (State). Each item is rated on a four point Likert scale. High STAI scores signify greater anxiety. The Trait anxiety is measured only once and the State at each time point.
► Economic outcomes will include costs to the NHS, quality-adjusted life years (QALYs) based on responses to the EQ-5D-5L, and the primary outcome, axillary recurrence. For more details, see Economic Analysis section below.
► Local (breast or chest wall) recurrence; defined as pathologically (cytology or biopsy) and/or radiologically confirmed recurrence after mastectomy in the skin or soft tissue of the chest wall within the anatomical area bounded by the mid-sternal line, the clavicle, the posterior axillary line and the costal margin or any type of breast carcinoma in the breast after conservation therapy.
► Regional (nodal) recurrence; defined as pathologically (cytology or biopsy) and/or radiologically confirmed recurrent tumour in the lymph nodes in the ipsilateral axilla, infraclavicular, supraclavicular fossa, interpectoral area or ipsilateral internal mammary chain.
► Distant metastasis; defined as confirmed metastasis (positive pathology and/or definitive evidence on

**Table 1** Secondary outcomes are assessed at the following time points

| Secondary outcome | Assessment time point (months) | | | | | | |
|---|---|---|---|---|---|---|---|
| | 3 | 6 | 12 | 24 | 36 | 48 | 60 |
| Arm morbidity | | X | X | X | X | | |
| Quality of life | X | X | X | X | X | | |
| Anxiety | X | X | X | X | X | | |
| NHS costs | X | X | X | X | X | X | X |
| EQ-5D-5L | X | X | X | X | X | | |
| Incremental cost per QALY gained | | | | | X | | |
| Incremental cost per reduction in axillary recurrence | | | | | | | X |
| Local (breast or chest wall) recurrence | | X | X | X | X | X | X |
| Regional (nodal) recurrence | | X | X | X | X | X | X |
| Distant metastasis | | X | X | X | X | X | X |
| Time to axillary recurrence | | X | X | X | X | X | X |
| Axillary recurrence free survival | | X | X | X | X | X | X |
| Disease free survival | | X | X | X | X | X | X |
| Overall survival | | X | X | X | X | X | X |
| Contralateral breast cancer | | X | X | X | X | X | X |
| Non-breast malignancy | | X | X | X | X | X | X |

QALY, quality-adjusted life year.

imaging) in all other sites of recurrence and may include those classified as: soft-tissue category, visceral category, central nervous system and skeletal spread.

► Time to axillary recurrence is the time between the date of randomisation and the date of axillary recurrence, measured in days.
► Axillary recurrence free survival is the time between the date of randomisation and date of confirmed axillary recurrence or date of death, whichever comes first, measured in days.
► Disease free survival; defined as the time between the date of randomisation and the date of disease progression (ie, local or regional recurrence or distant metastasis) or death, whichever comes first, measured in days.
► Overall survival is the time between the date of randomisation and the date of death from any cause.
► Contralateral breast cancer is new primary malignancy in the opposite breast unless obviously contiguous with recurrent chest wall disease or proven on cytology/biopsy to be of metastatic origin.
► Non-breast cancer is any new non-breast primary malignancy, except for adequately treated, superficial squamous or basal cell carcinoma of the skin, or carcinoma in situ of the cervix.

## Participant identification

Potential trial participants are identified at multidisciplinary meetings. A participant information leaflet and video is given to the participant

Based on local hospital practices, there are four participant pathways for recruitment:
1. After primary breast surgery: women who have a positive sentinel node are approached with regard to trial participation at the first postoperative clinic visit to offer trial participation.
2. Before primary breast surgery: in some hospitals sentinel node assessment is performed during primary breast surgery. In these hospitals, women are approached about the trial before surgery. If they are willing to participate consent is obtained prior to surgery. Randomisation is intraoperative if the SNs are confirmed as positive.
3. Post chemotherapy: women may be approached about the trial during and after chemotherapy but before starting axillary treatment (ANC or ART).
4. SNB (sentinel node biopsy) prior to neoadjuvant therapy: where the sentinel node biopsy is performed prior to neoadjuvant therapy patients can be approached during or after neoadjuvant therapy but prior to axillary treatment (ANC or ART).

### Study assessments
► Baseline: baseline data are collected from the participant's medical notes and questionnaires.
► Follow-up: follow-up assessments are at 6, 12, 24, 36, 48 and 60-month post randomisation with an additional postal questionnaire at 3 months.

The 6, 12, 24, 36, 48 and 60-month follow-up is in person or over the telephone, according to local practice. If contact cannot be made with the participant then medical notes can be used.

## Data collection

Trial data are entered at site into electronic case report forms (eCRF). Participants are identified only by their unique participant ID number. Access to the database is restricted and secure.

Data from routine medical notes can be used to collect outcomes for those who have been lost to follow-up or no longer wish to attend research visits.

## Monitoring of quality and safety

Data handling and trial monitoring procedures are detailed in trial-specific data management and trial monitoring plans. Data quality and compliance with the protocol is assessed throughout the trial by central monitoring and site visits where required. The trial will undergo periodic audits by the trial's unit quality assurance team and the trial sponsor.

All reportable Serious Adverse Events are sent directly to the trial's unit and assessed by the Chief Investigator and reported as part of annual reports.

## Confidentiality

Data obtained as a result of this study is considered confidential and confidentiality is ensured by utilising identification code numbers to correspond to treatment data.

## Flagging with NHS digital

Trial participation ends after 60 months with no further post-trial care (beyond standard practice). However, all women recruited to the study in the UK will be 'flagged' after discharge through the Data Linkage and Extract Service at NHS Digital to maximise outcome data collection and enable long-term follow-up data (greater than 5 years) to be reported.

## Statistics

### Sample size

Among women who have adjuvant therapy plus axillary treatment, 2% have axillary recurrence within 5 years.[11] A total of 1700 women is required for analysis (850 in each group) in order to detect a non-inferiority margin of two percentage points for the risk difference using the Miettinen-Nurminen score interval,[12] with 80% power and 2.5% one-sided alpha. Allowing for up to 10% non-compliance with treatment allocation and non-collection of primary outcome data, the target sample size to be randomised is 1900.

The magnitude of the non-inferiority margin was determined following consultation with relevant breast cancer consumer groups, lay people, breast surgeons and oncologists. There was consensus across all groups that an absolute increase in axillary recurrence of up to 2% is an acceptable trade-off for the increase in arm morbidity associated with axillary treatment.

### Statistical analysis

Analysis will be at the end of the 5-year follow-up of the last recruited participant. A detailed statistical analysis plan will be developed by the trial statisticians in consultation with the Trial Management Group, and agreed with the Trial Steering Committee before database lock and unblinding of data.

Descriptive statistics of demographic and clinical measures will be used to assess balance between the randomised arms at baseline. Continuous variables will be summarised in terms of the mean, SD, median, lower and upper quartiles, minimum, maximum and number of observations. Categorical variables will be summarised in terms of frequency counts and percentages.

The primary approach for between-group comparative analyses of primary and secondary endpoints will be per protocol, and we will present point estimates and 95% CIs. Sensitivity analysis where participants are analysed as randomised, regardless of compliance with allocation, will also be conducted for the primary outcome.

The primary analysis will assess whether adjuvant therapy alone is non-inferior to adjuvant therapy plus axillary treatment based on the primary endpoint of axillary recurrence within 5 years. The risk difference will be estimated using a mixed effects generalised linear model for binary outcome, with treatment arm and minimisation variables as fixed effects and recruitment site as a random effect. Non-inferiority of adjuvant therapy alone compared with adjuvant therapy plus axillary treatment will be accepted if the upper boundary of the Miettinen-Nurminen two-sided 95% confidence limit for estimated difference in axillary recurrence is no greater than 2%.

We will conduct subgroup analyses of axillary recurrence and arm morbidity by fitting appropriate interaction terms in the regression models, according to the following variables measured at baseline: number of sentinel node macrometastases; age; type of breast surgery; ER status; tumour grade; sentinel node assessment technique; extranodal invasion. We will present subgroup-specific effects and 95% CIs, as well as interactions, 95% CIs and p values. Due to the limited power available to detect all except large interactions, these subgroup analyses will be regarded as exploratory and interpreted with due caution.

Analyses for surgical complications following treatment of ANC will be of a descriptive nature.

## Economic analysis

The economic evaluation will be carried out from the perspective of the NHS over a 3-year and 5-year time horizon. A 3-year time horizon was adopted for the cost-utility analysis as it was anticipated, that by 3-years post randomisation, any differences in QoL between the intervention and standard care participants which might have been present earlier will have disappeared. Costs will be based on the costs of the randomised interventions received and on the use of subsequent healthcare and services.

For each participant, all clinical events (1) relevant to management and consequences following the initial sentinel node biopsy (adjuvant therapy alone or adjuvant therapy plus axillary treatment) (2) up to the end

of follow-up (3) which occur in the hospital setting, will be recorded. These include primary treatments, readmissions for complications or recurrences and associated adverse events. Events deemed unrelated to the consequences of the sentinel node biopsy or axillary treatment will not be included. In addition to these clinical events, participants will be asked to recall the extent to which they accessed primary care (eg, GP (general practioner) practice visits) or other non-hospital services (eg, physiotherapy visits) for their breast cancer at 6, 12, 24, 36, 48 and 60 months post randomisation. Information relating to healthcare resource use will be collected via the eCRF. Costs will be estimated using routine sources.[13 14] On the basis of each participant's record and the costs collected from routine sources, a total hospital cost and a total non-hospital cost will be derived for each participant for the duration of follow-up. These total costs will be combined to estimate the total cost per participant from which the total average cost per randomised group will be calculated. The breast and sentinel node biopsy procedure, systemic treatment, RT to breast or chest wall will not be included in the determination of costs, because all participants will undergo these treatments.

1. Cost-effectiveness analysis based on the incremental cost per reduction in axillary recurrence. Mean costs for each randomised arm will be calculated as will mean axillary recurrence and will be presented as point estimates of mean incremental costs and effects (reduced axillary recurrences) and the incremental cost per reduction in axillary recurrence at 5 years post randomisation.

2. Cost-utility analysis based on the incremental cost per QALY gained. QALYs will be based on the area under the curve approach[15] using responses to the EQ-5D-5L completed at scheduled time points; baseline, 3, 6, 12, 24, and 36 months post-randomisation. The total average QALYs accruing in the intervention and standard care groups will be analysed for significant differences, following allowances for mortality (if occurring). Both mean cost and QALYs will be presented for each randomised group and incremental mean costs and QALY calculated along with the incremental cost per QALY gained at 3 years post randomisation.

Both deterministic and stochastic sensitivity analyses will be used to address any uncertainty in our estimation of costs, effects and cost-effectiveness. Any uncertainty in the results, based on the stochastic sensitivity analyses, will be illustrated using cost-effectiveness planes and cost-effectiveness acceptability curves. Costs and effects will be discounted at UK recommended rates[16] as they are being estimated beyond a 1-year time horizon.

## Radiotherapy quality assurance

The UK Radiotherapy Trials Quality Assurance (RTTQA) Group will be responsible for implementing and coordinating the radiotherapy quality assurance (RT QA) programme through a named contact for the trial. The programme includes pre-trial and on-trial activities and is outlined below:

Pretrial QA—prior to site activation

► Facility questionnaire—Details of treatment technique, immobilisation, verification and dosimetry will be recorded.

► Dummy run—All centres must submit a test participant case to the RTTQA group, for review, demonstrating planned RT treatment to the breast/chest wall, axilla and supraclavicular fossa. The outlining, planning and treatment of this participant case should reflect how the centre intends to plan and treat all participants recruited to POSNOC.

On-trial QA—Ongoing QA during trial recruitment

► Retrospective individual case reviews —All centres must export RT plan data to the RTTQA Group for their first ten participants. Data will include participant history, CT data, structure set, plan and dose files. The retrospective review process should include at least three plans with RT to the nodal region.

► Ongoing data collection—RT data related to the RT QA programme will be collected and stored by the RTTQA Group for all participants, regardless of study allocation. A random sample of plans will be reviewed from each site during the trial recruitment period.

The sites will be required to adhere to the POSNOC RT planning and delivery guidelines when treating participants in the study. The RTTQA Group will promote and monitor compliance to the guidelines for the duration of trial recruitment by systematically reviewing a sample of RT plans submitted by centres. RT guideline deviations, non-compliance or areas of concern will be recorded and if required, discussed with the TMG and individual centres.

## Patient and public involvement

As part of the development of the project and protocol, women from the Derby Breast Cancer Support Group, Bosom Buddies (Guildford Breast Cancer Support Group) and Independent Cancer Patients' Voice (ICPV) have commented at all stages on the design and acceptability of the study. The Breast Cancer Care and Lymphoedema Support Network have had the opportunity to contribute to this protocol.

ICPV is an independent patient advocate group (http://independentcancerpatientsvoice.org.uk/). Two members of ICPV will provide a patient perspective throughout the duration of the study.

## Ethics and dissemination

The study has been approved by the National Research Ethics Service Committee East Midlands—Nottingham 2 (REC reference: 13/EM/0459).

The dissemination of the study results will be via a study report and research papers for publication in peer-reviewed journals, and presentation at relevant conferences. Reporting will be in compliance with Consolidated Standards of Reporting Trials recommendations.

Publication of the results will be based on outcomes at 5 years following the last recruited participant.

A summary of the results will be made available to participants through a newsletter (unless they state they do not wish to receive this), and will also be publicised through ICPV, Cancer Research UK, Breast Cancer Now and Lymphoedema Support Network.

**Author affiliations**
[1] Department of Surgery, University Hospitals of Derby and Burton NHS Foundation Trust, Derby, UK
[2] Department of Surgery, The University of Melbourne, Melbourne, Victoria, Australia
[3] Breast Cancer Trials, Newcastle, New South Wales, Australia
[4] Sussex Health Outcomes Research & Education in Cancer (SHORE-C), Brighton & Sussex Medical School, University of Sussex, Brighton, UK
[5] Nottingham Clinical Trials Unit, University of Nottingham, Nottingham, UK
[6] Brighton and Sussex Medical School, Brighton, UK
[7] Nuffield Department of Population Health, Oxford University, Oxford, UK
[8] Department of Oncology and Metabolism, Weston Park Hospital, Sheffield, UK
[9] Research and Development, University Hospitals of Derby and Burton NHS Foundation Trust, Derby, UK
[10] Department of Primary Care and Public Health, Cardiff University, Cardiff, UK
[11] School of Economics, University of Nottingham, Nottingham, UK
[12] Lymphoedema Department, University Hospitals of Derby and Burton NHS Foundation Trust, Derby, UK
[13] School of Medicine, University of Nottingham, Nottingham, UK
[14] Patient Advocate, Independent Cancer Patients' Voice, London, UK
[15] Health Economics Group, Population Health Sciences Institute, Newcastle University, Newcastle upon Tyne, UK
[16] National Radiotherapy Trials Quality Assurance Group, Mount Vernon Cancer Centre, Northwood, UK

**Acknowledgements** We thank the patients who are participating in this trial, and staff at the participating centres and the independent data monitoring committee and trial steering committee for overseeing the trial.

**POSNOC Trialists** POSNOC Trial Management Group: Trial Management Group members: Amit Goyal (Chief Investigator and Chair), Shabina Sadiq, Mickey Lewis, Rachel Haines, Alan Montgomery, Wei Tan, G Bruce Mann, Kathryn Monson, Valerie Jenkins, Patricia Fairbrother, David Dodwell, Tara Homer, Heath Badger, Zohal Nabi, Romaana Mir, Elizabeth Miles, Luke Vale, Malcolm W Reed, Lesley Fallowfield, Robert Coleman, Vaughan Keeley, Ian Ellis, Daniel Davis, Teresa Grieve (past members: Lelia Duley, Diane Whitham, Margaret Childs, Apostolos Fakis, Robert G Newcombe, David Whynes, Eleanor Mitchell, Rebecca Haydock, Clare Brittain, Clare Upton, Sarah Craig). Nottingham Clinical Trials Unit staff: Alan Montgomery, Rachel Haines, Shabina Sadiq, Wei Tan, Mickey Lewis, Aisha Shafayat, Charlotte Gidman, Mara Ozolins, Lelia Duley, Diane Whitham, Margaret Childs, Eleanor Mitchell, Rebecca Haydock, Clare Brittain, Clare Upton, Sarah Craig. Patient Reported Outcomes staff (Sussex Health Outcomes Research & Education in Cancer (SHORE-C)): Kathryn Monson, Valerie Jenkins, Lesley Fallowfield. Health Economics staff (Newcastle University): Tara Homer, Luke Vale (past staff: Laura Ternent). Breast Cancer Trials Australia and New Zealand staff: Heath Badger, Bruce Mann, Nicole Francis, Flonda Probert, Lisa Paksec, Rose Lucas (past staff: Annette Dempsey, Rochelle Thornton). Radiotherapy Quality Assurance team: Daniel Megias, Zohal Nabi, Preetinder Singh, Andrei Caraman, Elizabeth Miles, Romaana Mir, Roeum Butt, Hannah Price. Independent Trial Steering Committee members: Alastair Thompson (Chair), Peter Barrett Lee, Julie Wolfarth, Ian White, Nisha Sharma, Alistair Ring, Ramsey Cutress (past members: April Matthews).Independent Data Monitoring Committee members: Judith Bliss (chair), Daniel Rea, Peter Dubsky

**Contributors** Conceptualisation and writing original protocol: AG conceived the study and prepared the first draft of the original protocol. Funding acquisition (UK and ANZ): AG, GBM, LF, LD, MR, DD, REC, AF, RN, VJ, DW, MC, DW, VK, IE, PF. Trial design and methodology: all authors contributed to study design and methodology (AG, GBM, LF, LD, MR, DD, REC, AF, RN, VJ, DW, MC, DW, VK, IE, PF, SS, KM, AM, WT, LV, TH, HB, RHH, ML, DM, ZN, PS, AC, EM). Statistics: AF, RN, AM, WT. Review and editing: all authors read, critically revised and approved the manuscript before submission (AG, GBM, LF, LD, MR, DD, REC, AF, RN, VJ, DW, MC, DW, VK, IE, PF, SS, KM, AM, WT, LV, TH, HB, RHH, ML, DM, ZN, PS, AC, EM). Radiotherapy QA: DD, DM, ZN, PS, AC, EM. Trial management and co-ordination: all authors are current or past TMG members and contributed to trial management and co-ordination (AG, GBM, LF, LD, MR, DD, REC, AF, RN, VJ, DW, MC, DW, VK, IE, PF, SS, KM, AM, WT, LV, TH, HB, RHH, ML, DM, ZN, PS, AC, EM).

**Funding** The trial is sponsored by the University Hospitals of Derby and Burton NHS Foundation Trust (uhdb.sponsor@nhs.net). Sponsor staff - Daniel Davis, Joanne Thornhill, Anne Shaw, Amy Farmer, Teresa Grieve. This project is funded by the National Institute for Health Research Health Technology Assessment Programme (project number 12/35/17). Australian and New Zealand participation is funded by National Health and Medical Research Council (NHMRC) (project grant 1083172).

**Disclaimer** The views expressed are those of the authors and not necessarily those of the Health Technology Assessment Programme, NIHR, NHS or the Department of Health and Social Care.

**Competing interests** LV was a member of the NIHR Health Technology Assessment Programme, Clinical Trials and Evaluation Panel from 2014 to 2018.

**Patient and public involvement** Patients and/or the public were involved in the design, or conduct, or reporting, or dissemination plans of this research. Refer to the Methods section for further details.

**Patient consent for publication** Not applicable.

**Provenance and peer review** Not commissioned; externally peer reviewed.

**ORCID iDs**
Amit Goyal http://orcid.org/0000-0002-2381-8337
Rachel Helen Haines http://orcid.org/0000-0001-7924-0602

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
