## [Reviewer comments · BMJ Open]

ARTICLE DETAILS

TITLE (PROVISIONAL)	Protocol: POSNOC - Positive Sentinel Node: adjuvant therapy alone versus adjuvant therapy plus Clearance or axillary radiotherapy. A randomised controlled trial of axillary treatment in women with early stage breast cancer who have metastases in one or two sentinel nodes
AUTHORS	Goyal, Amit; Mann, G. Bruce; Fallowfield, Lesley; Duley, Lelia; Reed, Malcolm; Dodwell, David; Coleman, Robert E.; Fakis, Apostolos; Newcombe, Robert; Jenkins, Valerie; Whitham, Diane; Childs, Margo; Whynes, David; Keeley, Vaughn; Ellis, Ian; Fairbrother, Patricia; Sadiq, Shabina; Monson, Kathryn; Montgomery, Alan; Tan, Wei; Vale, Luke; Homer, Tara; Badger, Heath; Haines, Rachel; Lewis, Mickey; Megias, Daniel; Nabi, Zohal; Singh, Preetinder; Caraman, Andrei; Miles, Elizabeth

VERSION 1 – REVIEW

REVIEWER	Tam, Ka-Wai Taipei Medical University
REVIEW RETURNED	05-Jul-2021

GENERAL COMMENTS	The POSNOC trial is a highly rewarded study, which has been expected to complement evidence of previous ACOSOG-Z0011(Z11) trial on axillary management of early breast cancer. It evaluates whether for women with early breast cancer and one or two macrometastases, adjuvant therapy alone is non-inferior to adjuvant therapy plus axillary treatment (ANC or ART), in terms of 5-year axillary recurrence. The strengths of this study are extended population, large sample size, complete data of participants and time-dependent outcomes. However, the following issues still require attention. 1. The POSNOC includes women with T1 or T2, unifocal or multifocal invasive breast cancer, undergoing mastectomy or breast conserving surgery (BCS), one or two macrometastases at sentinel node biopsy, and with or without extranodal extension, which extended target population compared to the ACOSOG-Z0011(Z11) trial and provided a full-fledged evidence. However, the inclusion criteria of Z0011 trial regarding positive SLNs was less than three SLNs containing metastatic breast cancer documented under the premise of taking at least three SLNs during surgery. In POSNOC trial, patients with one or two macrometastases of SLNs were identified and included, but how many SLNs in total were taken during operation? It should be
---

noted that different proportions of positive SLN have different meanings on disease stage and prognosis, and omission of axillary management for all patients with one or two macrometastases of SLNs was debatable. The authors need to address it.

2. According to the inclusion criteria of POSNOC, patients with early breast cancer received sentinel node biopsy prior to neoadjuvant therapy were included and randomized to either intervention group or control group. As for these patients, did they receive sentinel node biopsy again during operation? If they did and were assigned to intervention group, but with macrometastases of SLNs found at the end, didn't they undergo any further axillary treatment with ANC or ART?

3. As mentioned in the protocol, allocation of the study is stratified by recruiting site and minimised by age, type of surgery, estrogen receptor status, number of positive nodes, and intra-operative sentinel assessment using OSNA. What is the reason for choosing "intra-operative sentinel assessment using OSNA" as one of the stratifications? The five stratifications for randomization seemed difficult to perform intra-operatively, please explain how to execute it.

4. In this study, the longest follow-up period was five year, which was rather sufficient and acceptable for evaluation of axillary recurrence and local recurrence. As for overall survival, disease-free survival, and other assessment of morbidity, further long-term follow-up was strongly recommended.

REVIEWER	Magnoni, Francesca European Institute of Oncology, Breast Cancer Surgery
REVIEW RETURNED	20-Jul-2021

GENERAL COMMENTS	-Pending the results of the 5-year outcome, the manuscript is presented as an updated report of the ongoing randomised, multicentre, non-inferiority POSNOC trial which recruitment started on 1 August 2014 and as of 09 June 2021, with 1866 participants randomised and an estimated enrolment of 1900. Primary endpoint is the 5 years axillary recurrence's rate in early breast cancer patients presenting one or two nodes with macrometastases undergone adjuvant therapy alone compared to adjuvant therapy plus axillary treatment. -Significant study in the current scientific perspective that aims to increasingly enhance an axillary conservative approach, in this case underlining the role of adjuvant therapies, given the growing prominence of tumor biology in the management of breast cancer. -Some of points of strength:  • it includes women undergoing mastectomy and those with extranodal invasion • it includes multi-focal invasive tumour ≤5 cm in its largest dimension -In the Introduction Authors underlined several limitations of Z0011 trial: "...Z11 has not changed clinical practice because of several limitations. The study did not meet its recruitment goal,
---

approximately 40% of participants had micrometastases, many were lost to follow-up (19.4%), and there was a lack of radiation therapy quality assurance. ...". These considerations would not seem so crucial in the overall premise of the scientific role of POSNOC trial in axillary management for breast cancer. It would be advisable to emphasize as a premise that the Z0011 finding "...are not applicable to subgroups that were not included such as those with mastectomy and microscopic extranodal invasion...", as written by Authors below.

- In relation to this sentence, the presence of microscopic extranodal invasions considered as an inclusion criteria could be clearly specified in the inclusion criteria in the web site <https://clinicaltrials.gov/ct2/show/NCT02401685>, in particular in case of selected cases of neoadjuvant therapy in patients with early breast cancer.

-Moreover, the statement "...Z11 does not change clinical practice..." seems rather open to criticism, given a large part of the literature demonstrating, on the contrary, the important and incontrovertible change in clinical practice offered by the Z0011 trial, both in terms of quality of life offered to women and in terms of costs reduction (Huang, Tzu-Wen et al. "Axillary Management in Women with Early Breast Cancer and Limited Sentinel Node Metastasis: A Systematic Review and Metaanalysis of Real-World Evidence in the Post-ACOSOG Z0011 Era." *Annals of surgical oncology* vol. 28,2 (2021): 920-929. doi:10.1245/s10434-020-08923-7/Morrow, Monica et al. "Surgeon Attitudes Toward the Omission of Axillary Dissection in Early Breast Cancer." *JAMA oncology* vol. 4,11 (2018): 1511-1516.

doi:10.1001/jamaoncol.2018.1908/ Mattar, Denise et al. "Economic implications of ACOSOG Z0011 trial application into clinical practice at the European Institute of Oncology." *European journal of surgical oncology : the journal of the European Society of Surgical Oncology and the British Association of Surgical Oncology*, S0748-7983(21)00580-1. 12 Jun. 2021, doi:10.1016/j.ejso.2021.06.016).

VERSION 1 – AUTHOR RESPONSE

Reviewer: 1

Dr. Ka-Wai Tam, Taipei Medical University

Comments to the Author:

1. The POSNOC includes women with T1 or T2, unifocal or multifocal invasive breast cancer, undergoing mastectomy or breast conserving surgery (BCS), one or two macrometastases at sentinel node biopsy, and with or without extranodal extension, which extended target population compared to the ACOSOG-Z0011(Z11) trial and provided a full-fledged evidence. However, the inclusion criteria of Z0011 trial regarding positive SLNs was less than three SLNs containing metastatic breast cancer documented under the premise of taking at least three SLNs during surgery. In POSNOC trial, patients with one or two macrometastases of SLNs were identified and included, but how many SLNs in total were taken during operation? It should be noted that different proportions of positive SLN have different meanings on disease stage and prognosis, and omission of axillary management for all patients with one or two macrometastases of SLNs was debatable. The authors need to address it.

The POSNOC protocol does not require the surgeons to remove at least 3 nodes during sentinel node biopsy. Surgeons are advised to perform sentinel node biopsy in a standard manner.

Z0011 trial did not require surgeons to remove at least 3 nodes. The median number of nodes removed in Z0011 study was 2 (IQR 1-4).

Giuliano AE, McCall L, Beitsch P, Whitworth PW, Blumencranz P, Leitch AM, Saha S, Hunt KK, Morrow M, Ballman K. Locoregional recurrence after sentinel lymph node dissection with or without axillary dissection in patients with sentinel lymph node metastases: the American College of Surgeons Oncology Group Z0011 randomized trial. *Ann Surg.* 2010 Sep;252(3):426-32).

2. According to the inclusion criteria of POSNOC, patients with early breast cancer received sentinel node biopsy prior to neoadjuvant therapy were included and randomized to either intervention group or control group. As for these patients, did they receive sentinel node biopsy again during operation? If they did and were assigned to intervention group, but with macrometastases of SLNs found at the end, didn't they undergo any further axillary treatment with ANC or ART?

These patients did not undergo sentinel node biopsy again after neoadjuvant chemotherapy.

3. As mentioned in the protocol, allocation of the study is stratified by recruiting site and minimised by age, type of surgery, estrogen receptor status, number of positive nodes, and intra-operative sentinel assessment using OSNA. What is the reason for choosing "intra-operative sentinel assessment using OSNA" as one of the stratifications? The five stratifications for randomization seemed difficult to perform intra-operatively, please explain how to execute it.

OSNA analyses and amplifies mRNA from solubilised biopsy samples of sentinel lymph node tissue. It detects the level of expression of the CK19 gene, an epithelial marker associated with breast cancer. There is more than 95% concordance between OSNA and histopathology, however, to minimise the imbalance between the trial arms, minimisation was used. Patients undergoing sentinel node assessment by OSNA were randomised intra-operatively.

4. In this study, the longest follow-up period was five year, which was rather sufficient and acceptable for evaluation of axillary recurrence and local recurrence. As for overall survival, disease-free survival, and other assessment of morbidity, further long-term follow-up was strongly recommended.

We agree with the reviewer. Long-term follow up is planned. This is reported on page 10 (section - Flagging with NHS Digital).

Reviewer: 2

Dr. Francesca Magnoni, European Institute of Oncology

Comments to the Author:

-In the Introduction Authors underlined several limitations of Z0011 trial: "...Z11 has not changed clinical practice because of several limitations. The study did not meet its recruitment goal, approximately 40% of participants had micrometastases, many were lost to follow-up (19.4%), and there was a lack of radiation therapy quality assurance. ...". These considerations would not seem so crucial in the overall premise of the scientific role of POSNOC trial in axillary management for breast cancer. It would be advisable to emphasize as a premise that the Z0011 finding "...are not applicable to subgroups that were not included such as those with mastectomy and microscopic extranodal invasion...", as written by Authors below.

- In relation to this sentence, the presence of microscopic extranodal invasions considered as an inclusion criteria could be clearly specified in the inclusion criteria in the web site <https://clinicaltrials.gov/ct2/show/NCT02401685>, in particular in case of selected cases of neoadjuvant therapy in patients with early breast cancer.

This has now been added to the text under target population on page 5. See below:

Target population: Women with unifocal or multifocal invasive breast cancer with the largest lesion ≤ 5 cm, who have 1 or 2 sentinel nodes with macrometastases (>2 mm), with or without extranodal invasion.

-Moreover, the statement "...Z11 does not change clinical practice..." seems rather open to criticism, given a large part of the literature demonstrating, on the contrary, the important and incontrovertible change in clinical practice offered by the Z0011 trial, both in terms of quality of life offered to women and in terms of costs reduction (Huang, Tzu-Wen et al. "Axillary Management in Women with Early Breast Cancer and Limited Sentinel Node Metastasis: A Systematic Review and Metaanalysis of Real-World Evidence in the Post-ACOSOG Z0011 Era." *Annals of surgical oncology* vol. 28,2 (2021): 920-929. doi:10.1245/s10434-020-08923-7/Morrow, Monica et al. "Surgeon Attitudes Toward the Omission of Axillary Dissection in Early Breast Cancer." *JAMA oncology* vol. 4,11 (2018): 1511-1516. doi:10.1001/jamaoncol.2018.1908/ Mattar, Denise et al. "Economic implications of ACOSOG Z0011 trial application into clinical practice at the European Institute of Oncology." *European journal of surgical oncology : the journal of the European Society of Surgical Oncology and the British Association of Surgical Oncology*, S0748-7983(21)00580-1. 12 Jun. 2021, doi:10.1016/j.ejso.2021.06.016).

The wording has been edited to below:

Z11 has not had the desired impact on change of clinical practice because of several limitations.

VERSION 2 – REVIEW

REVIEWER	Tam, Ka-Wai Taipei Medical University
REVIEW RETURNED	26-Sep-2021
GENERAL COMMENTS	All the comments have been addressed by the author.